# Tebentafusp in Patients with Metastatic Uveal Melanoma: A Real-Life Retrospective Multicenter Study

**DOI:** 10.3390/cancers15133430

**Published:** 2023-06-30

**Authors:** Dirk Tomsitz, Theresa Ruf, Markus Heppt, Ramon Staeger, Egle Ramelyte, Reinhard Dummer, Marlene Garzarolli, Friedegund Meier, Eileen Meier, Heike Richly, Tanja Gromke, Jens T. Siveke, Cindy Franklin, Kai-Christian Klespe, Cornelia Mauch, Teresa Kilian, Marlene Seegräber, Bastian Schilling, Lars E. French, Carola Berking, Lucie Heinzerling

**Affiliations:** 1Department of Dermatology and Allergy, University Hospital, Ludwig Maximilian University (LMU) Munich, 80337 Munich, Germany; dirk.tomsitz@med.uni-muenchen.de (D.T.);; 2Department of Dermatology, Uniklinikum Erlangen, Friedrich-Alexander-University Erlangen-Nürnberg (FAU), 91054 Erlangen, Germany; 3Comprehensive Cancer Center Erlangen-European Metropolitan Area of Nuremberg (CCC ER-EMN), 91054 Erlangen, Germany; 4Deutsches Zentrum für Immuntherapie (DZI), Friedrich-Alexander-University Erlangen-Nürnberg (FAU), 91054 Erlangen, Germany; 5Department of Dermatology, University Hospital Zurich, 8091 Zurich, Switzerland; 6Department of Dermatology, Faculty of Medicine and University Hospital Carl Gustav Carus, Technische Universität Dresden, 01307 Dresden, Germany; 7Skin Cancer Center at the University Cancer Centre Dresden and National Center for Tumor Diseases, 01309 Dresden, Germany; 8Department of Medical Oncology, West German Cancer Center, University Hospital Essen, 45147 Essen, Germany; 9Division of Solid Tumor Translational Oncology, German Cancer Research Center (DKFZ) and German Cancer Consortium (DKTK), Partner Site Essen, 69120 Heidelberg, Germany; 10Bridge Institute of Experimental Tumor Therapy, West German Cancer Center, University Hospital Essen, University of Duisburg-Essen, 45147 Essen, Germany; 11Department of Dermatology and Venereology, Faculty of Medicine and University Hospital Cologne, University of Cologne, 50937 Cologne, Germany; 12Center for Integrated Oncology Aachen-Bonn-Cologne-Düsseldorf (CIO ABCD), 50937 Cologne, Germany; 13Department of Dermatology, Allergology and Venereology, Hannover Medical School, 30625 Hannover, Germany; 14Department of Dermatology, University Hospital Würzburg, 97080 Würzburg, Germany; 15Dr. Philip Frost Department of Dermatology and Cutaneous Surgery, University of Miami Miller School of Medicine, Miami, FL 33136, USA

**Keywords:** uveal melanoma, tebentafusp, T cell engager, ImmTAC, real-life data, overall survival

## Abstract

**Simple Summary:**

Tebentafusp has recently been approved for the treatment of metastatic uveal melanoma (mUM). We performed a retrospective, multicenter study to analyze the outcomes and safety of tebentafusp therapy in 78 patients with mUM. Patients treated with tebentafusp had a median PFS of 3 months (95% CI 2.7 to 3.3) and a median OS of 22 months (95% CI 10.6 to 33.4). In contrast to a published Phase 3 study, our cohort had a higher rate of patients with elevated LDH (65.4% vs. 35.7%) and included patients with prior systemic and local ablative therapies. In patients treated with tebentafusp following ICI, there was a trend for a longer median OS (28 months, 95% CI 26.9 to 29.1) compared to the inverse treatment sequence (24 months, 95% CI 13.0 to 35.0, *p* = 0.257). The most common treatment-related adverse events were cytokine release syndrome in 71.2% of patients, which was managed with antipyretic drugs (66.1%), intravenous fluids (28.6%) and systemic corticosteroids or tocilizumab (5.4%), and skin toxicity in 53.8%, which was managed with topical corticosteroids (38.1%) or antihistamines (45.2%).

**Abstract:**

Background: Tebentafusp has recently been approved for the treatment of metastatic uveal melanoma (mUM) after proving to have survival benefits in a first-line setting. Patients and Methods: This retrospective, multicenter study analyzed the outcomes and safety of tebentafusp therapy in 78 patients with mUM. Results: Patients treated with tebentafusp had a median PFS of 3 months (95% CI 2.7 to 3.3) and a median OS of 22 months (95% CI 10.6 to 33.4). In contrast to a published Phase 3 study, our cohort had a higher rate of patients with elevated LDH (65.4% vs. 35.7%) and included patients with prior systemic and local ablative therapies. In patients treated with tebentafusp following ICI, there was a trend for a longer median OS (28 months, 95% CI 26.9 to 29.1) compared to the inverse treatment sequence (24 months, 95% CI 13.0 to 35.0, *p* = 0.257). The most common treatment-related adverse events were cytokine release syndrome in 71.2% and skin toxicity in 53.8% of patients. Tumor lysis syndrome occurred in one patient. Conclusions: Data from this real-life cohort showed a median PFS/OS similar to published Phase 3 trial data. Treatment with ICI followed by tebentafusp may result in longer PFS/OS compared to the inverse treatment sequence.

## 1. Introduction

Uveal melanoma (UM) is a rare intraocular tumor that is located either in the choroid (90%), ciliary body (6%) or iris (4%) [1]. Almost 50% of patients with UM eventually develop distant metastases, of which 89% can be found in the liver [2]. UM with epithelioid cell type, high mitotic activity and monosomy 3 are associated with metastatic disease and a poor prognosis [3]. Immune checkpoint inhibitors (ICI) ipilimumab and nivolumab, which have shown an outstanding median overall survival (OS) rate of 72.1 months in patients with metastatic cutaneous melanoma [4], are less effective in patients with metastatic UM (mUM). A lower mutational burden and lower programmed cell death ligand 1 expression in UM cells are assumed to make UM less immunogenic and hence less responsive to ICI in comparison to cutaneous melanoma [5]. Treatment with combined ipilimumab and nivolumab has led to a median PFS of 3.0 months and 5.5 months and a median OS of 12.7 months and 19.1 months in two small prospective Phase 2 trials with 52 and 33 patients, respectively [6,7]. A comparable median OS of 18.4 months was reached in nine patients with UM who were treated with a combination of locally ablative therapies and combined low-dose ipilimumab (1 mg/kg) and pembrolizumab (2 mg/kg) [8]. Interestingly, Grade 3/4 adverse events were observed in only 18% of these patients, compared to a rate of 59% of patients with cutaneous melanoma who were treated with ipilimumab (3 mg/kg) and nivolumab (1 mg/kg) [9]. As hepatic metastases occur frequently in patients with mUM, liver-directed therapies such as transarterial chemoembolization, selective internal radiation therapy or brachytherapy are often applied [10]. A retrospective study in 19 patients treated with a combination of liver-directed therapy and ICI revealed a significantly improved median OS of 22.5 months compared to 11.4 months in the control group [11]. In addition, UM harbors GNAQ or GNA11 mutations but lacks activating BRAF mutations—thus excluding targeted therapies with BRAF/MEK inhibitors [12].

Tebentafusp is the first drug that has been approved for the treatment of unresectable or metastatic UM by the FDA and EMA after showing a survival of 21.7 months compared to 16.0 months in the control group in a Phase 3 study. It belongs to a new class of bispecific fusion proteins—the so-called immune mobilizing monoclonal T-cell receptors against cancer (ImmTAC), or T cell engagers (TCEs) [13]. Tebentafusp is composed of an HLA-A*02:01-restricted T cell receptor that is specific for the gp100 peptide, with a higher affinity than natural T cell receptors. After binding to HLA-A*02:01-positive UM tumor cells, the fused anti-CD3 single-chain variable fragment recruits and activates T cells, leading to the lysis of UM cells. Tebentafusp use is restricted to HLA-A*02:01 patients—the latter being detected in approximately 50% of Caucasians [14]. 

In an open-label phase 3 trial, HLA-A*02:01-positive patients with mUM were either assigned to treatment with tebentafusp or to the control group, in which patients received either ipilimumab, pembrolizumab or dacarbazine as the investigator’s choice of treatment [15]. Patients were not allowed to receive combined immunotherapy. To be eligible for randomization into the trial, no prior systemic treatment or prior regional, liver-directed therapy was allowed. Patients with abnormal liver function tests were also not able to participate in the study. Out of the 378 patients included, 252 patients were treated with tebentafusp and 126 patients were treated with the investigator’s choice (82% pembrolizumab, 13% ipilimumab, 6% dacarbazine). Despite a very low objective response rate of 9% at 6 months, patients who were treated with tebentafusp had a significantly higher PFS (31%) compared to the control group (19%), and at 1 year, a significantly higher OS of 73% compared to 59%, respectively (hazard ratio for death, 0.51; 95% CI, 0.37–0.71; *p* < 0.001).

The most frequently observed treatment-related adverse events (TRAE) were either cytokine-related—including pyrexia (76%), chills (47%) and hypotension (38%)—or cutaneous side effects, including exanthema (83%), pruritus (69%) and erythema (23%).

While patients in prospective clinical trials form a homogenous population regarding comorbidities, baseline laboratory values and prior or concomitant therapies, data from real-life studies represent the actual, heterogeneous patient population more realistically. In this retrospective multicenter study, we present the clinical outcomes of 78 patients with mUM who were treated with tebentafusp in a real-life setting. Importantly, this cohort also included patients who received second-line tebentafusp after progression under therapy with ICI.

## 2. Patients and Methods

This retrospective multicenter study included all patients with mUM who received at least one dose of tebentafusp at any of the seven participating skin cancer centers in two countries (Cologne, Dresden, Erlangen, Essen, Munich and Wurzburg in Germany and Zurich in Switzerland). Altogether, 78 patients were identified. The baseline was defined as the date of the first tebentafusp treatment. Clinical data at the baseline were extracted from electronic medical records—including patient demographics, Eastern Cooperative Oncology Group (ECOG) performance status, tumor genotype (as assessed by next-generation sequencing), sites and number of organ systems affected by metastases, level of lactate dehydrogenase (LDH) and previous local and systemic antitumor therapies. 

During the treatment period, data on safety (premedication, incidence, severity, and management of adverse events) and efficacy (tumor response, overall survival) were collected. Adverse events assigned to cytokine release syndrome (CRS) were classified according to American Society for Transplantation and Cellular Therapy Consensus Grading (ASTCT, 2019) [16]. Other adverse events were graded under the terms of the Common Terminology Criteria of Adverse events (CTCAE) version 5.0.

Progression-free survival, overall survival and best overall response were assessed locally. Tumor assessment was performed according to RECIST (Response Evaluation Criteria in Solid Tumors) version 1.1.

Continuous data are presented as median or ranges and categorical data are presented as percentages. Continuous variables were compared using the unpaired Student’s *t*-test. Categorical variables were compared using Fisher’s exact test. Progression-free survival 1 (PFS on first-line treatment, PFS1), PFS2 (PFS on second-line treatment) and overall survival (OS) were compared between patients who were treated with first-line tebentafusp and patients who were treated with first-line immune checkpoint inhibitors (ICI). Log-rank tests were performed to compare PFS1, PFS2 and OS between the two groups. *p* values < 0.05 were considered statistically significant. Statistical analyses were conducted with SPSS Version 27 (IBM, Armonk, NY, USA). Diagrams were created using Excel 2016 (Microsoft, Redmond, WA, USA). 

The study was approved by the institutional review board of the medical faculty of the Munich University Hospital (reference number 20-1122) and was conducted in accordance with the principles of the Helsinki Declaration.

## 3. Results

### 3.1. Patients’ Characteristics

A total of 78 patients with mUM were included in this multicenter retrospective study and analyzed for PFS and OS after receiving tebentafusp as a first- or second-line therapy. Patients had a median age of 63 years (range 27–91 years) and a sex ratio of 1:1. In our study cohort, enucleation was only performed in 12 patients as a therapy for primary UM; data on histological type as well as the status of monosomy 3 were not available, as the analysis of monosomy 3 is not standard care at the involved centers. Mutations were not assessed specifically for the retrospective trial, but were collected if data was available. Mutations (including BAP1 mutations) were assessed via Next-Generation Sequencing in 37 patients. In 16 patients a mutation (BAP1, EIF1AX, GNAQ, GNA11 and/or SF3B1) was detected. The majority (51.3%) of patients were still on tebentafusp therapy at the time of data collection; 28.2% were deceased and 20.5% had stopped treatment with tebentafusp due to disease progression (16.7%) or side effects (3.8%). LDH at the beginning of treatment with tebentafusp was normal in 34.6% of patients, above the upper limit of normal (ULN) in 44.6% of patients and >2× ULN in 21.8% of patients. Distant metastases were exclusively hepatic in 43.6%, both hepatic and extrahepatic in 53.8%, and only extrahepatic in 2.6% of patients. Prior to tebentafusp, 35.9% of the patients had already been treated with liver-directed therapies and 51.3% had already received a systemic anti-tumor therapy, including combined ipilimumab and nivolumab, anti-PD1 monotherapy or chemotherapy (Table 1).

### 3.2. Outcome

Among the 78 tebentafusp treated patients, at least one radiologic examination after treatment start was available for 69 patients: five patients had died before the first radiologic examination and four patients were lost to follow-up. Of the 69 evaluable patients, the best overall response was partial remission in six patients (8.7%), stable disease in 19 patients (27.5%) and progressive disease in 44 patients (63.8%)—or 49 patients (66.2%), when including the five patients who died before the first staging. The median PFS was 3 months (95% CI 2.7 to 3.3) and the median OS was 22 months (95% CI 10.6 to 33.4, Figure 1). 

### 3.3. Safety

In our study, 88.5% (n = 69) of treated patients experienced Treatment-Related Adverse Events (TRAE) within 24 h of tebentafusp infusion. Cytokine release syndrome (CRS) was the most frequently reported side effect and occurred in 71.2% (n = 56) of patients. In 66.1% (n = 37), only pyrexia and chills occurred, and were classified as Grade 1 according to ASTCT Consensus Grading and treated symptomatically with antipyretic drugs. Grade 2 CRS (additional hypotension, manageable with intravenous fluids) was detected in 28.6% (n = 16) of the patients. Grade 3 CRS with hypotension, requiring systemic corticosteroids or tocilizumab, was observed in 5.4% (n = 3) of the patients. There was no grade 4 CRS in our cohort. 

Skin toxicity was observed in 53.8% (n = 42) of the patients and was of grade 1 (according to CTCAE v.5) in 59.5% (n = 25), grade 2 in 35.7% (n = 15) and grade 3 in 2.4% (n = 1) of patients. In one patient, a bullous drug reaction was observed. Topical therapy with corticosteroids was applied in 38.1% (n = 16) and symptomatic treatment for pruritus with antihistamines was given in 45.2% (n = 19). 

Nausea and vomiting were observed in 14.1% (n = 11) of the patients and managed with antiemetic drugs. Two patients suffered from abdominal pain, which had to be treated with opioids. 

In one patient, a tumor lysis syndrome occurred after the first dose of tebentafusp—including a decrease in prothrombin time and highly elevated LDH, C-reactive protein, uric acid and creatinine in serum. Treatment included intravenous liquids and rasburicase. 

To reduce the severity or prevent the onset of TRAE, premedication was administered in 34.6% of patients. Premedication encompassed antihistamines (96.3%, n = 26), antipyretics (70.4%, n = 19), intravenous liquids (11.1%, n = 3), systemic corticosteroids (7.4%, n = 2) and/or anti-emetics (2.7%, n = 1). 

TRAE that emerged more than 24 h after application of tebentafusp appeared in 16.6% (n = 13) of patients; these included vitiligo-like hypo- and depigmentation of the skin (MAH = melanoma associated hypopigmentation) in 11.5% (n = 9), leukotrichia (depigmentation of the hair) in 5.1% (n = 4), fatigue in 3.8% (n = 3) and pancreatitis in 2.6% (n = 2) of patients. 

In our patient cohort, three patients (3.8%) discontinued tebentafusp due to adverse events, and no treatment-related deaths were documented (Figure 2).

### 3.4. Treatment Sequence

To investigate the influence of the treatment sequence of tebentafusp and ICI therapy, patients who were treated with both were analyzed for PFS1, PFS2 and OS. In our cohort, 40 such patients were identified—of which nine were treated with first-line tebentafusp and second-line ICI, and 31 with first-line ICI and second-line tebentafusp (Table 2). 

Patients treated with second-line tebentafusp after ICI therapy showed a longer median PFS2 of 4 months (95% CI 2.9 to 5.1), compared to a median PFS1 of 3 months (95% CI 2.5 to 3.5) for patients treated with first-line tebentafusp (*p* = 0.111).

Patients who were treated with first-line ICI showed a longer median PFS1 of 5 months (95% CI 2.6 to 7.4), compared to a median PFS2 of 3 months (95% CI 1.5 to 4.5) for patients treated with first-line tebentafusp and second-line ICIs (*p* = 0.123). 

A comparison of OS between the two groups showed a longer median OS of 28 months (95% CI 26.9 to 29.1) first-line ICI therapy as compared to 24 months (95% CI 13.0 to 35.0) for first-line tebentafusp treatment (*p* = 0.257, Figure 3 and Figure 4). 

### 3.5. Local Liver-Directed Ablative Therapies

To investigate the impact of local ablative therapies on outcomes, we compared the OS of patients who were treated with any liver-directed treatment (n = 36) with patients who did not receive any local treatment (n = 42). Local therapies included selective internal radiation therapy (n = 16), brachytherapy (n = 6), stereotactic body radiation therapy (n = 4), chemosaturation (n = 4), transarterial chemoembolization (n = 3), radiofrequency ablation (n = 2) and microwave ablation (n = 1). Those patients showed a median OS of 22 months (95% CI 6.6 to 37.4) compared to a median 24 months (95% CI 10.5 to 37.5) for patients who had no local liver-directed therapy (*p* = 0.873, Figure 5).

## 4. Discussion

This real-life multicenter study investigated the outcome of tebentafusp therapy and shows a median PFS of 3.0 months and a median OS of 22.0 months in a cohort of 78 patients with metastatic UM. Compared to the data of the Phase 3 trial, Nathan et al. reported a similar median PFS of 3.3 months and median OS of 21.7 months [15]—even though our cohort had a larger proportion of patients with an elevated baseline LDH of 66% compared to 36%, which is associated with a worse clinical outcome [17]. It included patients regardless of preexisting comorbidities and/or abnormal laboratory values, which is especially relevant for safety analyses.

In addition to this, this is the first study to investigate sequential therapies, as in the Phase 3 trial no previous systemic or local tumor therapy was allowed apart from surgical metastasectomy. In the published Phase 2 trial, all patients had at least one prior systemic treatment before tebentafusp therapy [18]. Of those, 72% were treated with ICI, with a lower percentage of combined anti-CTLA4 and anti-PD1 of 24% compared to 100% in our cohort. The 1-year OS was 76% in patients with a best overall response of complete response, partial response or stable disease following ICI, and 60% in patients with a best overall response of progressive disease on prior ICI therapy.

In our patient cohort, 51% had a systemic treatment prior to the first application of tebentafusp and 27% had a local liver-directed treatment. Of those patients who were treated with ICI prior to or after therapy with tebentafusp, 95% were treated with combined ipilimumab and nivolumab and 5% were treated with anti-PD1-antibodies as a monotherapy. There was a trend for a longer median PFS1 in patients who were treated with first-line ICI, reaching 5.0 months—compared to 3.0 months for patients who were treated with first-line tebentafusp. Additionally, in patients treated with second-line tebentafusp, there was a trend towards a longer PFS2, at 4.0 months—compared to 3.0 months in patients who were treated with second-line ICI. Regarding survival, there was a trend towards a longer median survival of 28.0 months for patients who were treated with first-line ICI followed by tebentafusp, as compared to 24.0 months for patients treated with first-line tebentafusp followed by ICI therapy. In an analysis of 55 independent studies with 2682 patients with mUM, tebentafusp had a higher median OS of 22.4 months, compared to 15.7 months for patients who were treated with a combined immune checkpoint blockade [19].

A possible mechanism of improved survival for ICI followed by tebentafusp could be that after ICI-induced T cell expansion [20], a higher number of lymphocytes are available for mobilization to the tumor site by tebentafusp. For blinatumomab, a bispecific T cell engager such as tebentafusp—which targets CD19 instead of gp100 and is approved for the treatment of B cell acute lymphoblastic leukemia—a higher percentage of T cells at baseline was associated with an improved outcome [21]. In addition, in humanized mice that were engrafted with human B cell acute lymphoblastic leukemia, combined pembrolizumab and blinatumomab were more efficient than treatment with blinatumomab alone [22]. Interestingly, in contrast to the observed beneficial impact of local liver-directed therapies together with ICI treatment on the OS of mUM-patients [11], there was no meaningful difference in the OS in patients who were treated with any locally ablative therapy compared to patients who had no local liver-directed therapy.

TRAE that occurred within 24 h of tebentafusp application were observed in 88.5% of the patients in our cohort, of which 71.2% were attributed to cytokine release and 53.8% were skin-related. Compared to the data of the Phase 3 study, in 99% of the patients a treatment-related adverse event occurred, with 89% experiencing CRS and 83% a rash [15]. The lower frequency of documented adverse events in our cohort might be explained by the premedication used, consisting of either antihistamines and/or antipyretics in 35% of patients, and the less stringent documentation outside prospective studies. In one patient, tebentafusp induced a life-threatening tumor lysis syndrome—a rare side effect that was also observed in one patient among the 245 patients treated within the Phase 3 trial. In contrast to cytokine-related adverse events, which mainly occur during the first tebentafusp infusions, ICI can induce a broad spectrum of so-called immune-related adverse events that can even be elicited after the cessation of ICI treatment and are mainly managed with corticosteroids [23].

The main limitation of this study is its retrospective design, with an assessment of side effects documented only in the patients’ records. Diagnostic work-up and treatment decisions were made according to local standards. Thus, optional examinations such as mutation analyses were not performed in every patient.

## 5. Conclusions

Our retrospective study confirms the efficacy and safety of tebentafusp in patients with mUM in a real-life setting, with comparable response rates and overall survival. First-line treatment with ICI followed by tebentafusp was associated with longer PFS and OS than the inverse sequence and warrants further investigation.

## Figures and Tables

**Figure 1 cancers-15-03430-f001:**
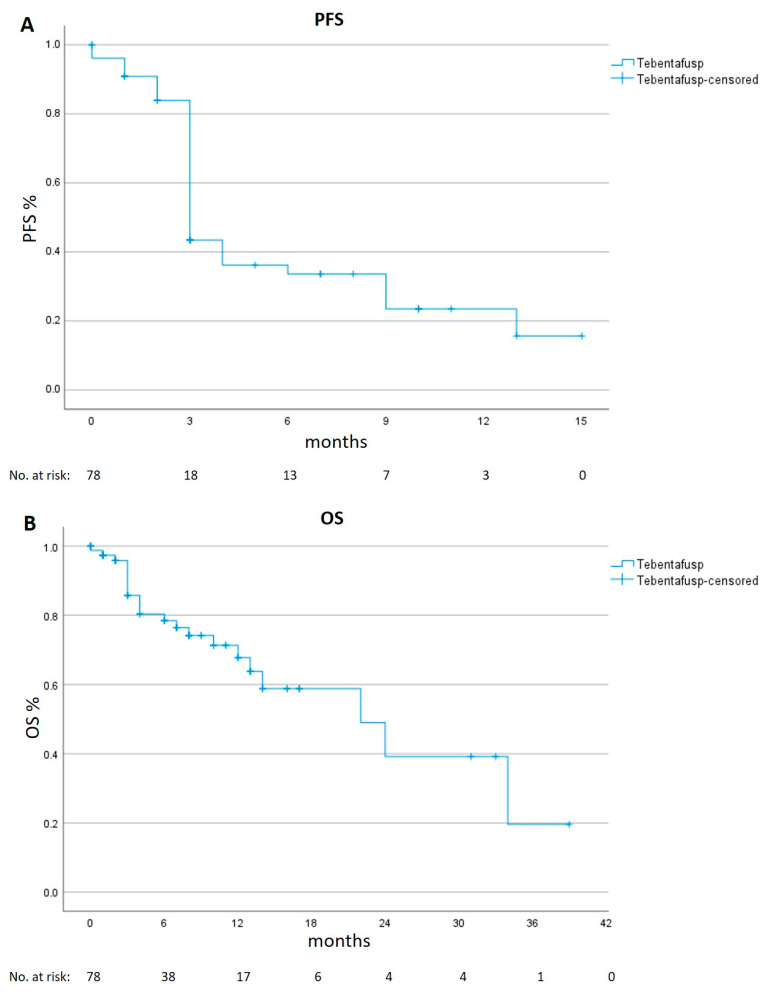
Progression-free (PFS) and Overall Survival (OS) in the real-life cohort. Kaplan–Meier estimates of PFS panel (**A**) and of OS panel (**B**).

**Figure 2 cancers-15-03430-f002:**
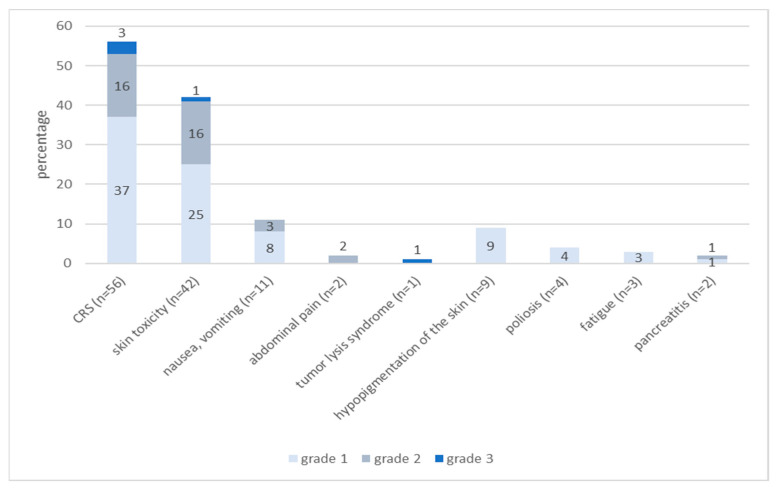
Frequencies and severity of Treatment-Related Adverse Events (TRAE). Cytokine release syndrome (CRS) was graded according to the 2019 recommendations of the American Society for Transplantation and Cellular Therapy (ASTCT) for consensus grading for CRS; other TRAE were graded according to CTCAE v.5. The occurrence of more than one TRAE was documented (values do not sum up to 100%). CRS = Cytokine release syndrome.

**Figure 3 cancers-15-03430-f003:**
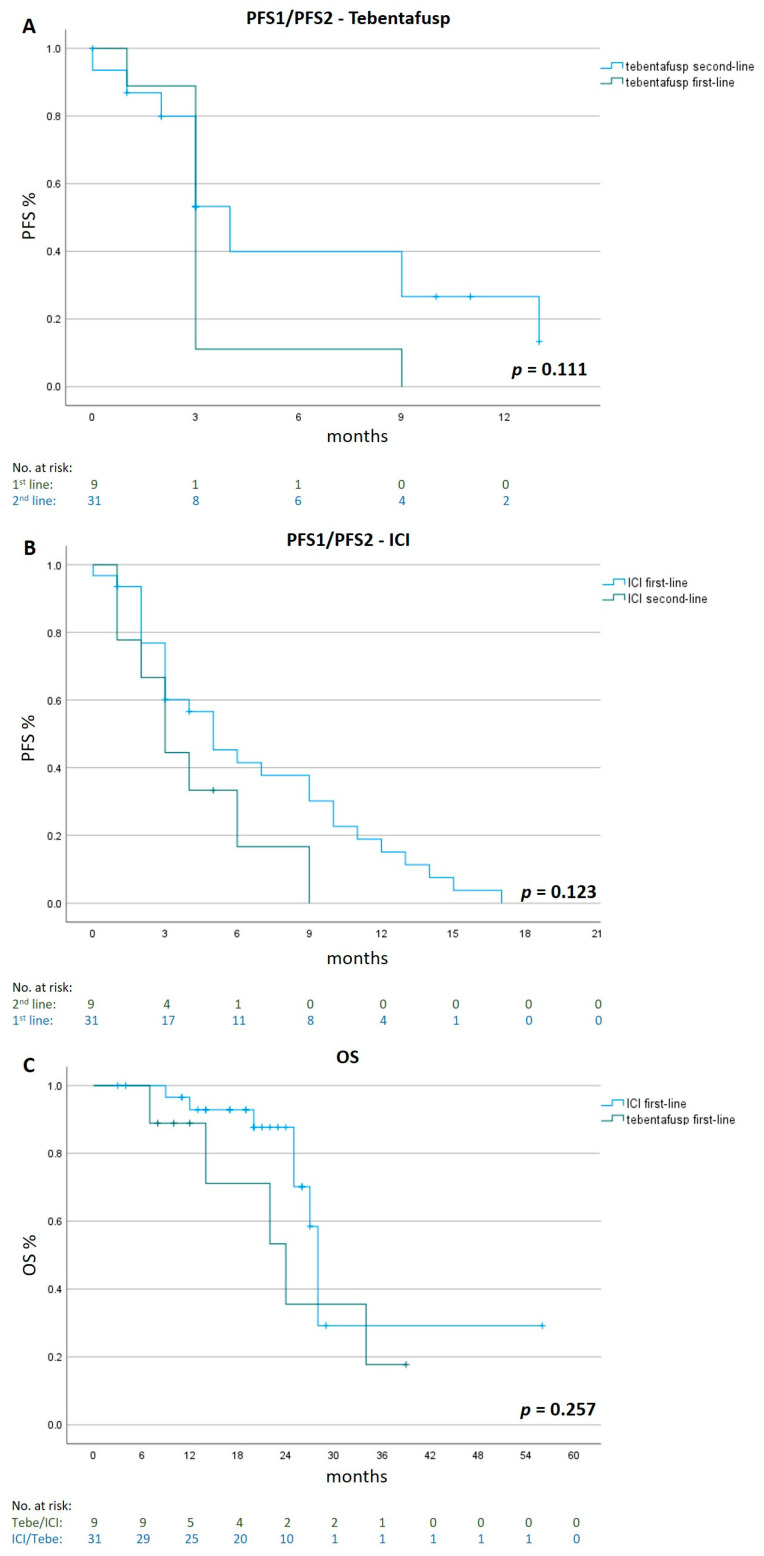
Progression-Free Survival 1/2 and Overall Survival in different treatment sequences. Kaplan–Meier estimates of PFS1/2 for tebentafusp given as first- (n = 9) and second-line (n = 31) therapy panel (**A**), PFS1/2 for ICI given in first (n = 31) and second-line (n = 9) panel (**B**) and of OS for patients treated with tebentafusp first-line (n = 9) or ICI first-line (n = 31) panel (**C**). ICI= immune checkpoint inhibitor.

**Figure 4 cancers-15-03430-f004:**
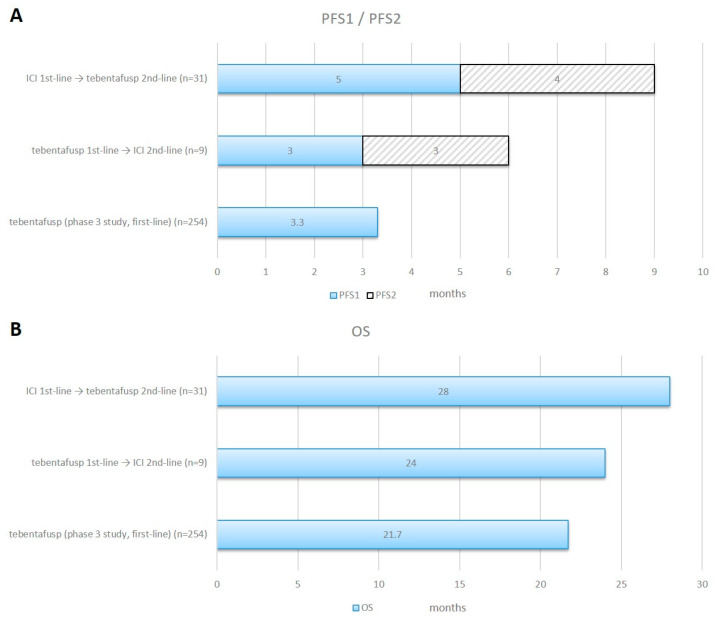
Progression-Free Survival 1/2 and Overall Survival in different treatment sequences. Horizontal bar charts of combined PFS1 and PFS2 panel (**A**) and OS panel (**B**) for ICI given in the first-line followed by tebentafusp in the second-line (n = 31) and tebentafusp in the first-line followed by ICI in the second-line (n = 9) compared with data from the Phase 3 study. ICI = immune checkpoint inhibitor.

**Figure 5 cancers-15-03430-f005:**
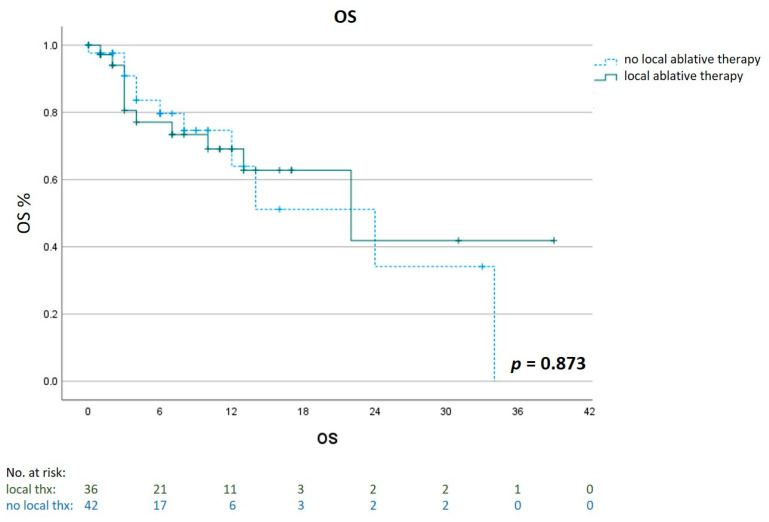
Impact of liver-directed therapies on Overall Survival (OS). Kaplan–Meier estimates of OS for patients who had an additional treatment with a liver-directed therapy (n = 36) compared to patients who had no local ablative treatment (n = 42).

**Table 1 cancers-15-03430-t001:** Baseline characteristics of the patients. ULN = upper limit of normal; ECOG = Eastern Cooperative Oncology Group; n/a = not available.

	Tebentafusp Real-Life Cohort (n = 78)	Tebentafusp Phase 3 Study (n = 252)
Age—yr		
median	63	64
range	27–91	23–92
sex—no. (%)		
male	39 (50.0)	128 (50.8)
female	39 (50.0)	124 (49.2)
treatment of primary UM		
external radiation therapy	27 (35.1)	
localized plaque radiation therapy	14 (17.9)	
enucleation	12 (15.4)	
watchful waiting	12 (15.4)	
theromotherapy	1 (1.3)	
unknown	12 (15.4)	
time between diagnosis of primary UM and distant metastases–months		
median	36.5	
range	0–195	
treatment status—no. (%)		
ongoing treatment with tebentafusp	40 (51.3)	n/a
stopped treatment with tebentafusp	16 (20.5)	n/a
dead	22 (28.2)	87 (34.5)
follow-up time after treatment with tebentafusp—months		
median	2	14.1
range	0–22	n/a
lactate dehydrogenase—no. (%)		
<ULN	27 (34.6)	162 (64.3)
>ULN	34 (44.6)	90 (35.7)
>2× ULN	17 (21.8)	n/a
ECOG performance status—no. (%)		
0	64 (82.1)	192 (76.2)
1	12 (15.4)	49 (19.4)
2	2 (2.6)	0 (0.0)
data missing	0 (0.0)	11 (4.4)
site of metastases—no. (%)		
hepatic only	34 (43.6)	131 (52.0)
hepatic and extrahepatic	42 (53.8)	111 (44.0)
extrahepatic only	2 (2.6)	9 (3.6)
data missing	0 (0.0)	1 (0.4)
number of metastatic sites—no. (%)		
1	35 (44.9)	n/a
2	19 (24.4)	n/a
3 and more	24 (30.8)	n/a
mutational status—no. (%)		
detected mutation	16 (20.1)	n/a
BAP1	7 (9.0)	n/a
EIF1AX	1 (1.3)	n/a
GNAQ	10 (12.8)	n/a
GNA11	7 (9.0)	n/a
SF3B1	5 (6.4)	n/a
no detection of mutation	21 (26.9)	n/a
unknown	40 (51.3)	n/a
preceding liver-directed therapy—no. (%)		
yes	21 (26.9)	0 (0.0)
no	57 (73.1)	252 (100.0)
preceding systemic tumor therapy—no. (%)		
yes	40 (51.3)	0 (0.0)
anti-PD1 antibodies	2 (2.6)	0 (0.0)
anti-PD1 + anti-CTLA4-antibodies	29 (37.2)	0 (0.0)
chemotherapy	9 (11.5)	0 (0.0)
no	38 (48.7)	252 (100.0)
subsequent liver-directed therapy—no. (%)		
yes	15 (19.2)	15 (6.0)
no	63 (80.8)	237 (94.0)
subsequent systemic tumor therapy—no. (%)		
yes	16 (20.5)	109 (43.3)
no	62 (79.5)	143 (56.7)

**Table 2 cancers-15-03430-t002:** Baseline characteristics of UM patients with more than one line of systemic therapy. ICI = immune checkpoint inhibitor, ULN = upper limit of normal, ECOG = Eastern Cooperative Oncology Group.

	First-Line Tebentafusp/Second-Line ICI (n = 9)	First-Line ICI/Second-Line Tebentafusp (n = 31)	*p*-Value
Age—yr			
median	65	60	0.418
range	43–75	49–78	
sex—no. (%)			
male	4 (44.4)	15 (48.4)	1.000
female	5 (55.6)	16 (51.6)	1.000
lactate dehydrogenase—no. (%)			
<ULN	4 (44.4)	8 (25.8)	0.411
>ULN	4 (44.4)	15 (48.4)	1.000
>2× ULN	1 (11.1)	8 (25.8)	0.654
ECOG performance status—no. (%)			
0	8 (88.9)	26 (83.9)	1.000
1	1 (11.1)	3 (9.7)	1.000
2	0 (0.0)	2 (6.5)	1.000
site of metastases—no. (%)			
hepatic only	7 (77.8)	12 (38.7)	0.060
hepatic and extrahepatic	2 (22.2)	19 (61.2)	0.060
skin	0 (0.0)	3 (9.7)	1.000
number of metastatic sites—no. (%)			
1	7 (77.8)	12 (38.7)	0.060
2	2 (22.2)	7 (22.6)	1.000
3 and more	0 (0.0)	13 (41.9)	0.019
immune checkpoint inhibitor—no. (%)			
ipilimumab/nivolumab	9 (100.0)	29 (93.5)	1.000
pembrolizumab	0 (0.0)	1 (3.2)	1.000
nivolumab	0 (0.0)	1 (3.2)	1.000

## Data Availability

All data generated or analyzed during this study are included in this article. Further enquiries can be directed to the corresponding author upon reasonable request.

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
