# Peer review of "Tebentafusp in Patients with Metastatic Uveal Melanoma: A Real-Life Retrospective Multicenter Study"

_cancers, 2023, doi:10.3390/cancers15133430_

Round 1

Reviewer 1 Report

Previous treatment deviation: The study includes patients who had already received various treatments prior to receiving tebentafusp. This can impact treatment response and make it challenging to attribute the results solely to tebentafusp itself.

I would advise the authors to provide more detailed specification of the pre-existing conditions of the patients to ensure a more homogeneous analysis. Additionally, adding more information about the patient selection process for the study would be beneficial.

Lack of discussion on study limitations: The article inadequately addresses the limitations and biases inherent in a retrospective study, which could influence the interpretation of the results.

The authors mentioned the use of SPSS for data analysis, but some of the graphs were created using Excel. I would recommend that the authors improve the quality of the tables.

Reviewer 2 Report

Overall, the manuscript is clearly written and represents some contribution to the field. Moreover, the methodology was apparently properly used and the study is rigorously executed, novel and highly clinically relevant. I have only a few suggestions/questions.

1)     This is a paper on Metastatic uveal melanoma, but there is no information at all on the original primary disease, uveal melanoma. How long has the patient had uveal melanoma, has it been treated with enucleation or other therapies? What is the histological type? What is the status of monosomy 3?  And other information about primary uveal melanoma?

2)     What are the side effects and efficacy of tebentafusp compared to ipilimumab and/or nivoumab, please include a literature discussion, even if it is speculation in the DISCUSSION. After all, it is hard for the reader to know which is better.

The manuscript is clearly written.

Reviewer 3 Report

I believe that the topic of research based on multicenter studies is needed in view of the controversy and the lack of effective methods of treating UM metastases.

Reviewer 4 Report

The manuscript entitled “Tebentafusp in patients with metastatic uveal melanoma: a 2 real-life retrospective multicenter study” reports on the effects of Tebentafusp on metastatic uveal melanoma. The paper compares the results of this real world study with those of the original trial and adds information on potential effects of the treatment sequence between Tebentafusp and immune checkpoint inhibitors. The highly timely study is important and well described, the results might impact clinical decision making on the treatment sequence.

Minor remarks:

The simple summary should contain a statement concerning the management of toxicity.

The authors should indicate how mutations were assessed, especially so for BAP1 mutations.

Round 2

Reviewer 1 Report

With the implemented improvements, the manuscript could be accepted for publication.